# Mechanical Constraint Effect on DNA Persistence Length

**DOI:** 10.3390/molecules27227769

**Published:** 2022-11-11

**Authors:** Cheng-Yin Zhang, Neng-Hui Zhang

**Affiliations:** 1Department of Mechanics, School of Mechanics and Engineering Science, Shanghai University, Shanghai 200444, China; 2Shanghai Key Laboratory of Mechanics in Energy Engineering, Shanghai Institute of Applied Mathematics and Mechanics, Shanghai University, Shanghai 200072, China

**Keywords:** double-stranded DNA, persistence length, buckling length, continuum model, mechanical constraint effect

## Abstract

Persistence length is a significant criterion to characterize the semi-flexibility of DNA molecules. The mechanical constraints applied on DNA chains in new single-molecule experiments play a complex role in measuring DNA persistence length; however, there is a difficulty in quantitatively characterizing the mechanical constraint effects due to their complex interactions with electrostatic repulsions and thermal fluctuations. In this work, the classical buckling theory of Euler beam and Manning’s statistical theories of electrostatic force and thermal fluctuation force are combined for an isolated DNA fragment to formulate a quantitative model, which interprets the relationship between DNA persistence length and critical buckling length. Moreover, this relationship is further applied to identify the mechanical constraints in different DNA experiments by fitting the effective length factors of buckled fragments. Then, the mechanical constraint effects on DNA persistence lengths are explored. A good agreement among the results by theoretical models, previous experiments, and present molecular dynamics simulations demonstrates that the new superposition relationship including three constraint-dependent terms can effectively characterize changes in DNA persistence lengths with environmental conditions, and the strong constraint-environment coupling term dominates the significant changes of persistence lengths; via fitting effective length factors, the weakest mechanical constraints on DNAs in bulk experiments and stronger constraints on DNAs in single-molecule experiments are identified, respectively. Moreover, the consideration of DNA buckling provides a new perspective to examine the bendability of short-length DNA.

## 1. Introduction

The semi-flexible mechanical properties of DNA play an important role in many biological processes, such as genome storage, transcriptional regulation, and viral infection [1,2,3,4]. Moreover, as one of the stiffest natural polymers, DNA is an ideal engineering material for nanofabrication because of its exquisite specificity of Watson–Crick base pairing [5]. So, a better understanding and manipulation of DNA mechanical properties is necessary to explore life and facilitate DNA nanoengineering.

The persistence length *l*_p_ is one important criterion for characterizing the local rigidity but global flexibility of semi-flexible DNA chains [2,6,7]. In statistical physics, the persistence length is defined as the sum of average projections of each segment vector in a sufficiently long DNA chain on the first segment vector [6], in simple terms, a DNA chain with contour length *l*_c_ >> *l*_p_ can be considered as a flexible strand, in contrast, that with *l*_c_ ≤ *l*_p_ can be considered an elastic rod [3]. The persistence length of double-strand DNA (dsDNA, and DNA mentioned below means dsDNA unless special instructions) is frequently approximated in the literature with about 50 nm at standard conditions [3,7].

The first measurement of DNA persistence length quantity was conducted in the 1950s [8]; in fact, the persistence length is not measured directly but extracted from original quantitative measurements by theoretical models [9]. Environmental conditions [10,11], intercalators [12,13], experimental methods [9], and theoretical models [10,14] will affect the values of DNA persistence length. One of the distinctive features of DNA persistence length is environmental dependence, and traditional bulk techniques, including light scattering [15,16], birefringence [17], cyclization [11], etc., revealed ionic and temperature dependences of DNA persistence length. In these bulk experiments, long DNA is suspended in solution. Recently, some research revealed that intermediate- and short-scale DNA also has significant biological functions, such as nucleosomes [18] and binding proteins [19]. The single-molecule experiments enable researchers to expand their understanding of DNA properties, especially for intermediate- and short-scale DNA [12,20].

In various single-molecule experiments, there are divergent mechanical constraints on DNA molecules. In the optical tweezers experiment, the force-extension curves can be obtained by stretching a DNA chain, and then the DNA persistence length can be extracted by the worm-like chain (WLC) model [12,20]. However, in order to fulfill the loading and deformation measurement, one end of the DNA chain should be anchored to a surface, while the other end should be attached to a micro bead held in an optical trap [10,21], obviously, the constraints of DNA chain in optical tweezers experiment are different from those of DNA chains suspended in bulk solution experiments. Furthermore, the optical tweezers experiment reveals that DNA persistence length increases with its contour length, which is not reflected in the WLC model, so it was speculated that the WLC model ignored some effects important to experiments [21]. While in the experiment of tethered-particle-motion (TPM), one end of the DNA chain is fixed and the other end is attached to a bead fluctuating with the flow field [22]. These constraints from fixed end and bead in optical tweezers and TPM experiments may lead to fewer conformations of DNA [23]. Interestingly, the TPM experiment also found the dependence of DNA persistence length on contour length [9]. In addition, atomic force microscopy (AFM) [13,24,25] and electron microscopy (EM) [26] are used to image shapes of DNA adsorbed on the substrate, by which the persistence length can be extracted. During the process of DNA adsorption on the substrate, the losses of degree of freedom and potential distortions for DNA occur, which may change its apparent persistence length. For example, in the EM experiment of Faas et al. [27], the persistence lengths of DNAs adsorbed on carbon film were identified as 25–30 nm, much less than the commonly accepted length of 50 nm; as well as in the AFM experiment of Manteli et al. [14], the persistence length of DNA adsorbed on a mica under higher ionic concentration was about half that determined by optical tweezers experiment. Therefore, the developing experimental techniques urgently require more simple and reasonable theories to interpret the blowout experimental results with higher accuracy and more details.

For the interpretation of ionic dependence of DNA persistence length, we use to resorting to the classical and modified Odijk-Skolnick-Fixman (OSF) models [28,29,30,31,32]. In these OSF models, DNA persistence length is characterized as the sum of two terms, the bare persistence length *l*_0_ and electrostatic one *l*_e_, i.e., *l*_p_ = *l*_0_ + *l*_e_, where bare term *l*_0_ is a constant and the electrostatic term decreases with increasing ionic strength [28,29,30,31,32], but OSF models do not reproduce the concave shape of DNA persistence length with ionic strength observed in experiments [9]. Podgornik et al. considered that segments of a charged chain are stretched by Debye–Hückel repulsion, and proposed coupling elastic modulus with the renormalized stretching modulus due to electrostatic interactions to interpret the dependence of elastic modulus of polyelectrolyte on ionic conditions [33]. Manning proposed a concept of the null isomer of DNA, which is a hypothetical uncharged DNA structure, and based on the buckling of the null isomer he derived an alternative nonadditive relationship between a new invariable term and DNA persistence length [34], and this model fits well data of DNA persistence length for Na^+^ salt, but it also stirs up the puzzle that the originally assumed invariable term varies with the counterion valency [9]. In addition, as mentioned above, the classical WLC model cannot be used to explain some results of single-molecule experiments, so we have to seek the following two potential directions for analyzing single-molecule experimental results. The first direction is based on statistical mechanics. As for the optical tweezers experiment, accounting for three features of mechanical constraint, namely the effects of finite chain length, chain-end boundary conditions, and bead rotational fluctuations, on DNA probability distribution, Seol et al. developed a finite worm-like chain (FWLC) model, and their predictions showed that mechanical end-constraints of DNA will affect the force-extension response of stretched DNA and the relevant DNA persistence length prominently [21]. As for the TPM experiment, Segall et al. used an entropic force to describe interactions between the fixed substrate and the tethered bead; therefore, the possible DNA conformations were analyzed, and the results showed that the larger tethered bead leads to more DNA motion and more conformations, implying that the constrained effect can alter DNA apparent properties [23]. Kurzthaler established exact Gibbs free energy for semi-flexible polymers subjected to external forces at ends and provided the configurations with clamped, cantilevered, and free boundary conditions, and the results showed that polymers subjected to compressive forces had S-shaped or hooked-shaped configurations for different boundary conditions, respectively [35]. The second direction is based on fitting parameters. Brunet et al. proposed an empirical equation of DNA persistence length with four parameters and fitted these parameters from TPM experimental results; the fitted results can better reflect the ionic dependence of DNA persistence length [9]. Wiggins et al. considered a DNA chain as several segments and proposed a sub-elastic chain (SEC) model, when a small length of a segment and a suitable parameter are given, the prediction for 2D distribution of DNA by the SEC model agreed well with the results by AFM experiments, and this suggests that properties of DNA under specific conditions should be analyzed concretely rather than just simply extrapolated from long-length-scale measurements [25]. To sum up, simple and reasonable theoretical models considering constraint effects are required to interpret new experimental results.

This paper is devoted to presenting a quantitative relationship between DNA persistence length and buckling length of bent DNA fragments under complex interactions among mechanical constraints, electrostatic repulsions, and thermal fluctuations. First, from the view of continuum mechanics, the deformation of constrained DNA fragment is viewed as beam buckling under the combined deformation state of bending and compression, and Euler’s model of continuum beam and Manning’s statistical models of electrostatic and thermal fluctuation forces are combined to characterize deformation of DNA fragment; therefore, a new quantitative relationship between DNA persistence length and critical buckling length is proposed. Second, this relationship is applied to fit the effective length factors from experimental results to identify mechanical constraints on DNAs in different experiments. Third, the mechanical constraint effects on DNA persistence lengths and their environmental dependence are investigated; and by comparing with the OSF models, the new additivity of this relationship for DNA persistence length contributed from the combined effects of electrostatic, thermal, and mechanical constraint is demonstrated, and some relevant mechanisms are revealed. Furthermore, the theoretical predictions are also verified by our molecular dynamics (MD) simulations.

## 2. Model and Methods

### 2.1. Relationship between DNA Persistence Length and Critical Buckling Length

Biological shape is revealing of the physical forces acting on it [36,37], as DNA conformation does. The negatively charged DNA is in a shape caused by the coupled effects of internal electrostatic repulsion, thermal fluctuations, and external mechanical constraints. When the environment fluctuates, these effects also change, resulting in shifts between different DNA states, which is displayed as a transition between DNA straight or bending shapes. This change between different shapes is similar to the change in the deformation states of a buckled elastic rod under axial compression [38]. Therefore, the buckling model of the Euler beam can be conveniently and effectively used to study DNA.

As the study object, a bending DNA fragment in a critical buckling state is isolated from a long DNA chain as shown in Figure 1a, so the length of the isolated fragment can be regarded as the critical buckling length *l*_cb_ under its end-constraint condition. Under the combined effect of mechanical constraints from adjacent DNA fragments at both ends, interior electrostatic repulsion caused by linearly distributed negative charges on DNA phosphate backbone and DNA thermal fluctuation at environmental temperature, the isolated DNA fragment will change its original straight state (blue dashed DNA in Figure 1a) to bending state. Here, for simplicity, the mechanical constraints at each end are modeled as a rotational spring and a lateral spring, and axial displacement is not allowed, as shown in Figure 1b. When the DNA fragment is buckled, the mechanical end-constraints generate support reaction forces and moments to balance the interior electrostatic repulsion and thermal fluctuation force in the axial direction.

The part of axial support reaction force contributed by electrostatic repulsion is derived from electrostatic free energy *G*_ele_: if the electrostatic change makes the DNA fragment in a small change of axial tension deformation Δ*L*, the resultant axial support reaction force *F*_ele_ will be compressive and does a negative work for tension deformation, i.e., *W* = −*F*Δ*L*. The change of electrostatic free energy is given as Δ*G*_ele_ = *G*_ele_ (*L* + Δ*L*) − *G*_ele_ (*L*), based on the reciprocal work theorem, one can get *W* = Δ*G*_ele_, then the axial support reaction force *F*_ele_ can be obtained by the following limit process with Δ*L* → 0
(1)$$F_{\mathrm{ele}}≜\underset{\Delta L\to 0}{\lim}\left(-\frac{G_{\mathrm{ele}}(L+\Delta L)-G_{\mathrm{ele}}(L)}{\Delta L}\right)=-\frac{\partial G_{\mathrm{ele}}(L)}{\partial L}$$

Here, Manning’s electrostatic free energy for the linearly charged polymer chain [39,40] is used and the support reaction force contributed by electrostatic repulsion is expressed as [34]
(2)$$F_{\mathrm{ele}}=\frac{k_{B}T}{Z^{2}l_{B}}\left[\left(2Z\xi -1\right)\frac{\kappa be^{-\kappa b}}{1-e^{-\kappa b}}-1-\ln\left(1-e^{-\kappa b}\right)\right]$$
where *k*_B_ is Boltzmann constant, *T* is environmental temperature, *Z* is unsigned valence of counterions, *l*_B_ = *q*^2^/4*πε*_0_*ε*_r_*k*_B_*T* is Bjerrum length [41], *q* is the elementary charge, *ε*_0_ is the absolute dielectric constant, *ε*_r_ is the relative dielectric constant, *ξ* = *l*_B_/*b* is a dimensionless polyion charge density and determines a contribution of electrostatic free energy from counterions condensation [39,40], *b* is the characterized spacing of charge arrayed on the DNA molecule, *κ* = 1/*λ*_D_ and *λ*_D_ = [(*ε*_0_*ε*_r_*k*_B_*T*)/(*cN*_A_*q*^2^*Z*^2^)]^1/2^ is the Debye length [42], *c* is the ionic concentration, and *N*_A_ is Avogadro’s constant.

In addition, another part of the axial support reaction force contributed by thermal fluctuation for DNA fragments under mechanical constraints is expressed as [34]
(3)$$F_{\mathrm{tf}}=\sqrt{EAk_{B}T/l_{\mathrm{cb}}}$$
where *E* is axial Young’s modulus of DNA, *A = πR*^2^ is area of circular DNA cross-section, and *R* is the radius of DNA cross-section. 

In Appendix A, the relationship between the axial compressive load and critical buckling length of a buckled DNA fragment with elastic end-constraints as shown in Figure 1b is derived. Though derivation in Appendix A, the elastic end-constraints can be described with a parameter *μ* named effective length factor, which is defined as the ratio of the critical buckling length with hinge-hinge end-constraint to that with any elastic end-constraint [43]. When the compressive load is substituted with *F*_tf_ + *F*_ele_, the critical buckling length *l*_cb_ of the fragment is expressed as
(4)$$l_{\mathrm{cb}}=\frac{\pi }{\mu }\sqrt{\frac{EI}{F_{\mathrm{tf}}+F_{\mathrm{ele}}}}$$
where *I* = *πR*^4^/4 is inertia moment of DNA cross-section. *EI* is DNA flexural rigidity, which has an alternative expression in the WLC model related to persistence length *l*_p_ as [7]
(5)$$EI=k_{B}Tl_{p}$$

Substitute Equations (3) and (5) into Equation (4), after expanding and eliminating the square root, Equation (4) can be transformed as
(6)$${\left(\frac{F_{\mathrm{ele}}^{}}{k_{B}T}\right)}^{2}l_{\mathrm{cb}}^{4}-\frac{4l_{p}}{R^{2}}l_{\mathrm{cb}}^{3}-\frac{2\pi^{2}l_{p}}{\mu }\left(\frac{F_{\mathrm{ele}}^{}}{k_{B}T}\right)l_{\mathrm{cb}}^{2}+\frac{2\pi^{4}l_{p}^{2}}{\mu^{4}}=0$$
which is quartic in critical buckling length *l*_cb_ and quadratic in persistence length *l*_p_. Solving Equation (6) for persistence length *l*_p_, a complex relationship between DNA persistence length and critical buckling length can be obtained as
(7)$$l_{p}=\frac{1}{\pi^{4}R^{2}}\left[2\mu^{4}l_{\mathrm{cb}}^{3}+2\mu^{3}l_{\mathrm{cb}}^{5/2}\sqrt{\pi^{2}R^{2}\frac{F_{\mathrm{ele}}}{k_{B}T}+\mu^{2}l_{\mathrm{cb}}}+\pi^{2}R^{2}\mu^{2}l_{\mathrm{cb}}^{2}\frac{F_{\mathrm{ele}}}{k_{B}T}\right]$$

In Equation (7), ionic dependence is introduced from force *F*_ele_, temperature dependence is introduced from thermal energy *k*_B_*T*, and constrained effect is introduced from effective length factor *μ*. Note that the competitive relationship between ionic and temperature effects of persistence length is revealed in the coupling term of *F*_ele_/*k*_B_*T* in Equation (7).

Note that if there is no electrostatic repulsion in DNA, namely force *F*_ele_ = 0, and the specific constraints at DNA fragment ends are simplified as clamp-clamp type, namely *μ* = 0.5, Equation (7) will degenerate into the relationship between persistence length *l*_p_^*^ and critical buckling length *l*_cb_^*^ of “null isomer” proposed by Manning, expressed as [34]
(8)$$l_{p}^{*}=\frac{l_{\mathrm{cb}}^{*3}}{4\pi^{4}R^{2}}$$

On the other hand, if thermal fluctuations can be neglected, namely force *F*_tf_ = 0, Equation (7) will degenerate into the previous expression of DNA buckling caused by a hypothetical opposite electrostatic tension force proposed by Manning, expressed as [34]
(9)$$l_{p}=\frac{l_{\mathrm{cb}}^{2}\mu^{2}}{\pi^{2}}\frac{F_{\mathrm{ele}}}{k_{B}T}$$

Note that our expression of DNA persistence length in Equation (7) is also similar in form to the classical OSF model [31,32]. Actually, the OSF model is a two-term superposition of electrostatic and non-electrostatic parts, whereas our model is a three-term superposition with one pure mechanical constraint term and two constraint-dependent environment terms.

### 2.2. Molecular Dynamics Simulation

In order to validate our continuum model, an all-atomistic MD simulation is performed to study the dependence of DNA persistence length on ionic and temperature conditions. Our MD simulation is inspired by the work of Wu et al., who used MD simulation to study the finite-length effects of flexibility of short DNA [44]. To balance the reliability and efficiency of MD simulation, we selected a short B-DNA with a contour length of 50 bp, the sequences of one single strand of DNA is 5’-CGACTCGACTCTACGGAAGGGCATCCTTCGGGCATCACTACGCGCCGCGC-3′, the other strand is completely complementary, which is the same as that in Wu’s MD simulation [44]. The DNA is immersed in a rectangle box (9.1 nm × 9.1 nm × 19.2 nm) containing water molecules and ions. For the study of ionic dependence, environmental conditions are set as 298 K temperature and three concentrations of NaCl solutions, 0.01 M, 0.1 M, and 1 M, respectively; for the study of temperature effect, environmental conditions are set as 0.1 M NaCl and three temperatures, 278 K, 298 K, and 313K, respectively, in all simulations pressure is set as 1 bar. In simulations, the Amber99SB-ILDN force field [45] and TIP4P water model are used. All systems are optimized, equilibrated, and run with the software Gromacs 5.1.4 [46]. After sequentially performing with energy minimization, temperature equilibration, and NPT equilibration, each simulation is run for 50 ns with time-step 2 fs, and one frame of DNA conformation is extracted from the simulated trajectory every nanosecond, so 50 DNA conformations are obtained for each condition. Afterwards, determining the central axes of DNA conformation helices by program Curves+ [47], we obtained the contour length and end-to-end distance of DNA from central axes so as to determine the persistence length by [2]
(10)$$\langle h_{e\text{-}e}^{2}\rangle =2l_{p}L_{\mathrm{DNA}}\left[1-l_{p}\left(1-e^{-L_{\mathrm{DNA}}/l_{p}}\right)/L_{\mathrm{DNA}}\right]$$
where $\langle h_{e\text{-}e}^{2}\rangle$ is the mean square end-to-end distance and *L*_DNA_ is the contour length.

## 3. Results and Discussion

In this section, first, the effective length factors and critical buckling lengths of DNA containing mechanical constraints in different experiments are obtained by fitting the reported experimental results with the present theoretical model. Second, the effects of ionic conditions, temperature, and constraints on DNA persistence length are investigated, and the reliability of our model is verified by comparing it with experimental results [9,10,15,16,17,48,49], MD results [50,51], and predictions by other theoretical models [31,32,34]. In the Discussion section, we elaborate on a potential mechanism of DNA bending conformation possibly caused by DNA buckling and also discuss the limitations of our model. In computation, Boltzmann constant *k*_B_ = 1.38 × 10^−23^ J/K, elementary charge *q* = 1.6 × 10^−19^ C, absolute dielectric constant *ε*_0_ = 8.85 × 10^−12^ F/m, relative dielectric constant *ε*_r_ is dependent on temperature, and *ε*_r_ = 87.74 − 0.4(*T* − 293) from 278 K to 313 K [52], characterized spacing of charge arrayed on DNA molecule *b* = 0.17 nm [34], Avogadro’s constant *N*_A_ = 6.022 × 10^23^, radius of DNA cross-section *R* = 1 nm.

### 3.1. Identify Effective Mechanical Constraints in Different Experiments 

Several experimental data [9,10,15,16,17,48,49] of DNA persistence length versus ionic concentration are used to fit effective length factor *μ* and critical buckling length *l*_cb_ of DNA with the present model Equation (7). The comparisons between theoretical fitting curves and corresponding experimental results are shown in Appendix A, and the present fittings curves can reflect well the obvious concave shape of DNA persistence length with ionic concentration [9]. Table 1 lists the details of these experiments and fitting results, including the experimental method, actual constraints on DNA, temperature, kinds and concentrations of salt solutions, DNA contour lengths, the fitted length factors, critical buckling lengths, and goodness of fit. Here we used the coefficient of determination, *r*^2^, for quantifying the goodness of fit to describe how well our model fits a set of experimental measurements, *r*^2^ is expressed as *r*^2^ = 1 − SSR/TSS, where SSR means the sum of squares of residuals and TSS is the total sum of squares. A goodness of fit more closer to 1 indicates that the model fits the experimental measurements better.

First, as shown in Table 1, the effective mechanical constraints at DNA fragment ends in different experiments are identified by curve fitting. The fitted value of *μ* corresponding to the state of DNA suspended in solution is close to 1, which is the effective length factor of a compressed rod with hinge-hinge end-constraint. This end-constraint implies that the adjacent DNA fragments can hardly restrict the rotation of isolated DNA fragments.

Second, the fitted value of *μ* corresponding to the state of DNA with fix-bead constraint closes to 0.7, which is the effective length factor of a compressed rod with hinge-clamp end-constraint, implying that the mechanical constraint caused by the fixed end and attached bead are stronger than that of DNA suspended in solution. In addition, for TPM experimental data [9], the fitted parameters *μ* or *l*_cb_ under different counterion valent conditions are similar, respectively, which means that effective length factor and critical buckling length are mainly dependent on actual mechanical constraints on DNA. In contrast, the fitted values of the persistence length of DNA null isomer, in Manning’s model have large divergences under different counterion valent conditions, which seems a slight deviation from his original assumption that the persistence length of DNA null isomer was not dependent on ionic conditions [9].

Third, the fitted value of *μ* corresponding to the state of DNA adsorbed on the substrate is close to 0.5, which is the effective length factor of a compressed rod with clamp-clamp end-constraints. However, this fitting has a lower goodness of fit, so the reliability of its effective mechanical constraint needs to be further verified, and the reasons for its low goodness of fit will be analyzed in the Discussion section. 

### 3.2. End-Constraint Effect on Ionic Dependence of DNA Persistence Length

In this section, the mechanical end-constraint effects on ionic dependence of DNA persistence length will be investigated. Ionic conditions are set as monovalent salt with a concentration range from 0.001 M to 4 M to cover the experimental range. Due to the low goodness of fit for DNA adsorbed on the substrate, this section only focuses on DNAs suspended in solution and with fix-bead constraint. The average effective length factors and critical buckling lengths and their standard deviations for the two constrained DNA states are obtained from corresponding data in Table 1, and listed in Table 2. 

By substituting the averaged values of critical buckling lengths *l*_cb_, effective length factors *μ*, and standard deviation of *μ* under different constrained states in Table 2 into Equation (7) with temperature *T* = 298 K, we can predict DNA persistence length versus monovalent ionic concentration as shown in Figure 2, and the error bars in Figure 2 is caused by the standard deviation of averaged effective length factors *μ*. Regardless of mechanical constraints, the DNA persistence length decreases with ionic concentration overall, which is consistent with reported experimental results [9,10,15,16,17,49] and theoretical predictions [28,31,34]. In addition, the ionic dependence of DNA persistence length is attributed to force *F*_ele_ in Equation (7) contributed by the electrostatic effect. It should be noted that the standard deviation of average critical buckling length is not involved in calculations to avoid repeatedly considering statistical errors; otherwise, the range of error bars in Figure 2 will be excessively large.

Another feature reflected in Figure 2 is that the DNA persistence length for DNA with fix-bead constraint is smaller than that of DNA suspended in solution. The previous statistical studies show that the mechanical constraint effects have close relation with the influence of constraint on DNA entropy [23,53]. While in Figure 2, the lower effective length factor indicates that the constraint of fix-bead is stronger than that of DNA suspended in solution, and the stronger constraint may restrict the number of possible DNA conformations, leading to lower DNA entropy and larger extension [21], which means more flexible DNA with smaller persistence length [14,21,25]. In addition, in Figure 2, the narrower distribution in the experimental results of DNA with fix-bead constraint and shorter error bars in the present predictions may also reflect a fewer number of possible DNA conformations.

Figure 3 shows the comparison of our predictions by the present model, our MD simulation results, and reported MD results [50,51] for the persistence length of DNA suspended in solution. As shown in Figure 3, all predictions for DNA persistence length decrease with ionic concentrations. It should be noted that Appendix A show DNA conformations in MD simulation from 0 to 50 ns every 10 ns, and Appendix A show the trends of DNA persistence length in our MD simulations having the convergence after 20 ns, so 30 sets of data by MD simulations from 20 ns to 50 ns are used to obtain the average DNA persistence length at each environmental condition in Appendix A. As shown in Figure 3, the present MD results are on the same order of magnitude as our model predictions and reported MD results, and all they have a consistent trend with ionic concentration, which further verifies the reliability of the present model.

Figure 4 shows the respective contributions of three additive terms in Equation (7) to DNA persistence length under different ionic concentrations by three colors in the histogram, and these results are also compared with the predictions by the additive OSF model [31,32] and multiplicative Manning’s model [34]. The first term only includes the effective length factor *μ* and buckling length *l*_cb_ related to constraints, so it can be called “term only depending on constraint (TODC)”; the second term is changed mildly with environmental conditions due to the square root of *F*_ele_/*k*_B_*T*, so the second term is called “term weakly depending on environment (TWDE)”; the third term is changed obviously with environmental conditions, so it is called “term strongly depending on environment (TSDE)”.

As shown in Figure 4, in a low range of ionic concentration (0.01–0.1 M), the TSDE in the present model can reveal a continuous change of DNA persistence length, whereas that in the OSF model changes more rapidly, which might lead to the inability of OSF model to reflect the concave shape of DNA persistence length varying with salt concentration [9]. On the other hand, in a high range of ionic concentration (>0.1 M), the TODC and TWDC can reveal a slow change of DNA persistence length, whereas that in Manning’s model still maintains a faster decline. In summary, the complex coupling effect among mechanical constraint, electrostatic, and temperature displayed in terms of the superposition relationship of TODC, TWDE, and TSDE makes the present model have a good prediction for DNA persistence length.

### 3.3. End-Constraint Effect on Temperature Dependence of DNA Persistence Length

In this section, we will investigate the end-constraint effect on the temperature dependence of DNA persistence length in the temperature range from 278 to 313 K, in which DNA does not undergo denaturation [54]. The cyclization experiment in Mg^2+^ solution performed by Geggier et al. revealed that DNA persistence length is almost linearly temperature-dependent expressed as an empirical expression *l*_p_ = (3.19 − 0.00414*T*) × 10^−19^/*k*_B_*T* nm [11]. In their experiment, DNA is a short-circularized molecule (~200 bp). Considering the short DNA contour length and cyclized conformation, DNA fragments may be symmetrically constrained at both ends, so the equivalent DNA end-constraints in Geggier’s experiment are assumed to be clamp-clamp type.

First, based on the clamp-clamp end-constraint of DNA in Geggier’s experiment, the effective length factor *μ* = 0.5 and temperature-dependent DNA persistence length obtained from Geggier’s empirical expression are substituted into Equation (6) to obtain the critical buckling length *l*_cb_ of DNA fragment with temperature, and the prediction is shown as a black dot in Figure 5. Obviously, the critical buckling length decreases approximately linearly with the increasing temperature, so a linear fit for critical buckling length with temperature can be expressed as
(11)$$l_{\mathrm{cb}}=\left(24.831-0.0359T\right)\times 10^{-9}$$

Second, it is assumed that the gradient of buckling length varying with temperature in Equation (11), i.e., −0.0359 nm/K, is also applicable for other mechanical end-constraints. The average critical buckling lengths 5.95 nm and 8.43 nm at 298 K listed in Table 2 and the obtained gradient are used to obtain the linear expression of DNA critical buckling lengths varying with temperature for DNA suspended in solution or with fix-bead constraint, respectively, i.e.,
(12)$$l_{\mathrm{cb}\text{-}\mathrm{suspend}}=\left(16.65-0.0359T\right)\times 10^{-9}$$
(13)$$l_{\mathrm{cb}\text{-}\mathrm{fix}\text{-}\mathrm{bead}}=\left(19.13-0.0359T\right)\times 10^{-9}$$

Therefore, with the help of Equations (12) and (13), the critical buckling lengths varying with temperature for DNAs with two end-constraint conditions are shown in Figure 5, obviously the increasing temperature results in a decrease in critical buckling length.

Third, by substituting Equations (12) and (13) into Equation (7), one can obtain the variations of the persistence length of DNA with different end-constraints in 0.1 M Na^+^ solution versus temperature, and Figure 6 shows the compassion of these predictions and our MD simulation for DNA suspended in solution. All the present predictions of DNA persistence lengths with different end-constraints decrease approximately linearly with increasing temperature, which reproduces a similar trend in Geggier’s experiment [11]. But different end-constraints cause different decreasing rates; the persistence length for DNA suspended in solution decreases more significantly than that with fix-bead constraint. The decreasing trend and magnitude in our MD simulations are consistent with the prediction for DNA suspended in solution. This further verifies the present model and also suggests the necessity of identifying mechanical constraints in DNA experiments by effective length factor.

Next, the changes of three additive terms in Equation (7) varying with temperature are also shown in Figure 7. Obviously, the TSDE dominates the temperature dependence of DNA persistence length, with a dramatic descending trend, especially for DNA suspended in solution.

## 4. Discussion

The first to be discussed is the potential relevance between DNA buckling and short-length DNA bending. There is an active controversy about enhanced DNA bendability at short lengths [55]. For example, fluorescence resonance energy transfer and X-ray scattering experiments found that DNAs of 3–30 nm lengths had spontaneous bending with 1.6–6.8 nm radius of gyration [56]; AFM experiments found that DNA of 10 nm length had significant bending [25]; molecular vise experiments found that DNA longer than 14 nm had buckling bending [57]; MD simulation results indicated that DNA of 24 nm length had C-shaped or S-shaped bending [38]. Interestingly note that the DNA critical buckling lengths listed in Table 1 are about 5–15 nm, which reproduces well with the above experiment and simulation results of enhanced DNA bendability at lengths shorter than persistence lengths. In addition, this also echoes the importance of the mechanical constraint effect revealed in the recent upgrading explorations of WLC models [21,35]. Therefore, we strongly believe that the enhanced DNA bendability at short lengths is closely related to DNA buckling, which is actually a shape demonstration of the selected DNA particle ensembles under the compensating physical forces including electrostatic force, thermal fluctuation force and mechanical constraint force [36,37].

The next to be discussed are the possible reasons for the lower goodness of fit for the EM experimental result [48] in Table 1. The first reason is that the adsorption on substrate exerts a distributed lateral effect on whole DNA in EM experiment, rather than just end-constraint. Some published research can be referred to the study of substrate effect on DNA. Wilber simplified the interaction between a graphene sheet and a flat rigid substrate as van der Waals force, and revealed how the sheet buckling depended on boundary conditions [58]; Abu-Salih et al. simplified the elastic foundation supporting a compressive-stressed infinite beam as a lateral distributed force, and revealed that the post-buckling deflection of this beam was periodic sinusoidal [59]; Wang et al. simplified nanochannel confinement on a semi-flexible polymer chain as a quadratic potential, and revealed that the confined effect could be effectively equivalent to a stretching force depending on the size and shape of nanochannel [60]. These studies may inspire the development of a model for DNA absorbed on substrate. The second reason is that there are different experimental values of the persistence length measured from DNA adsorbed on substrate [14,27,48]. Faas et al. and Manteli et al. ascribed this diversity to two possible extreme states of DNA adsorption on substrate: DNA equilibrating on substrate before its adsorption (called 2D state) or DNA captured by substrate without equilibration (called 3D state) [14,27]. However, in practice, it seems hard to identify real adsorption states with the current techniques and select an appropriate theoretical model to extract the correct persistence length [13,14].

The last to be discussed is that DNA persistence lengths by MD simulation are then theoretical predictions as shown in Figure 4 and Figure 6. The fitted parameters used in the prediction are based on experimental results for long DNA chains with tens of kbp contour length, whereas shorter DNA chains are used in present and reported MD simulations [50,51]. While it has been reported that shorter DNA contour length leads to smaller persistence length [21,25,44,61]. 

## 5. Conclusions

In this work, we treat the bending deformation of a constrained DNA fragment isolated from a long DNA chain by buckling rod theory in macroscopic continuum mechanics, and use mesoscopic models of electrostatic repulsion and thermal fluctuation on DNA [34] to describe the force causing deformation. In this way, an alternative multiscale quantitative model is established, this simple model can be effectively fitted to measured data from the literature [9,10,15,16,17,48,49] over a wide range of experimental conditions, including 0.002–4 M monovalent ionic buffer solution, the contour length of long DNA chains from 1200–39,000 bp with two experimental methods. The fitting parameters by a new three-term superposition relationship in this model can identify the effective mechanical end-constraints on DNAs in different experiments, then the mechanisms for the mechanical constraint effect on DNA persistence length and its environmental dependences are clarified. The predictions of DNA persistence length varying with ionic conditions, temperature, and constraints by the present theoretical model are in agreement with the previous experimental results [9,10,15,16,17,48,49], and the present and published MD results [50,51]. The related conclusions are as follows:

The new three-term superposition relationship similar to the OSF model can reveal the complex interactions among mechanical constraint, electrostatic repulsion, and temperature, which determines the flexibility of DNA persistence length varying with environmental conditions. The strong constraint-environment coupling term dominates a significant change of persistence length with environmental conditions, while the weak constraint–environment coupling term and the pure mechanical constraint term compensatingly maintain the stability of DNA persistence length, especially at extreme conditions. So, the three-term additive relationship provides a more accurate characterization of DNA persistence length in complex environments.

The effective mechanical end-constraints on DNAs are different in various typical bulk and single-molecule experiments revealed by fitting effective length factors with the present model. The average effective length factor for bulk DNA solution experiment is 1.021 ± 0.250 which means the corresponding constraint is approximately equivalent to hinge-hinge end-constraint, and that for the fix-bead constraint in experiments by optical tweezers or TPM is 0.718 ± 0.075 which means the corresponding constraint approximately equivalent to hinge-clamp end-constraint, while the constraint in cyclization experiment is approximately equivalent to clamp-clamp end-constraint. The results suggest that stronger local or global constraints will lead to smaller persistence length globally, which may be attributed to the reduction in mechanical constraints on DNA conformation entropy [21,23]. 

This study not only clarifies the mechanism of the divergence of DNA persistence length in various experiments but also suggests the importance of identifying mechanical constraints accompanied by new experimental techniques for measuring DNA properties. Moreover, the consideration of DNA buckling may provide a new perspective to explore the bendability of short-length DNA. However, this coarse-grained multiscale model cannot yet reflect the effects of base pair or DNA sequence, it would be improved with more refined models including more interaction information of microscopic particles to study.

## Figures and Tables

**Figure 1 molecules-27-07769-f001:**
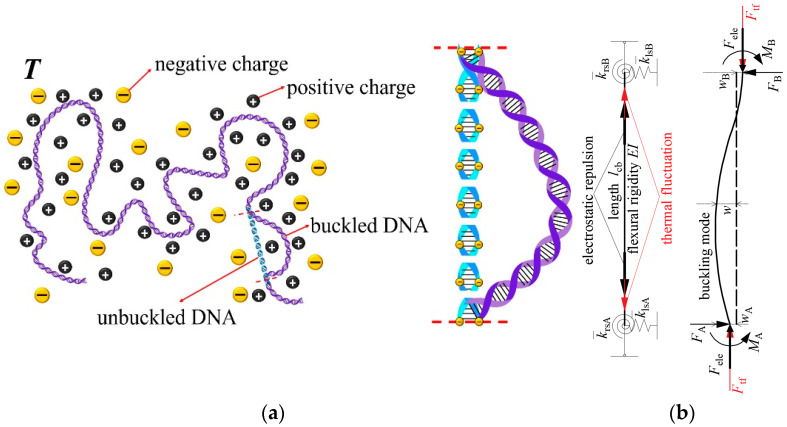
(**a**) A bent fragment isolated from a long DNA chain and (**b**) buckling of DNA fragment under the combined effects of interior electrostatic repulsion, thermal fluctuation, and effective mechanical end-constraints.

**Figure 2 molecules-27-07769-f002:**
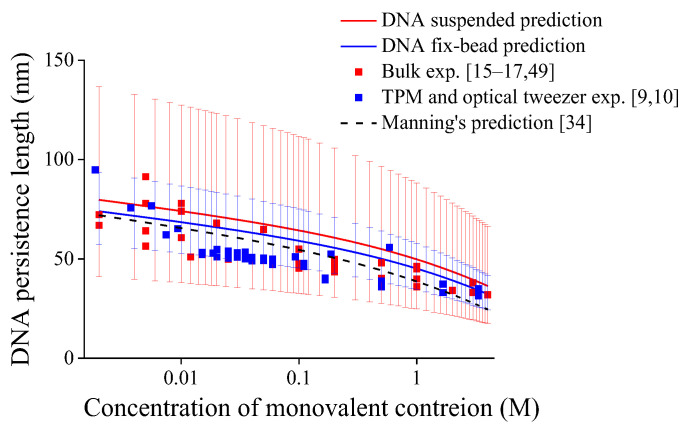
Persistence lengths of DNA under different end-constraints versus monovalent ionic concentrations, error bars are caused by standard deviations of averaged effective length factors with *μ* = 1.021 ± 0.250 for DNA suspended in solution while *μ* = 0.718 ± 0.075 for fix-bead one.

**Figure 3 molecules-27-07769-f003:**
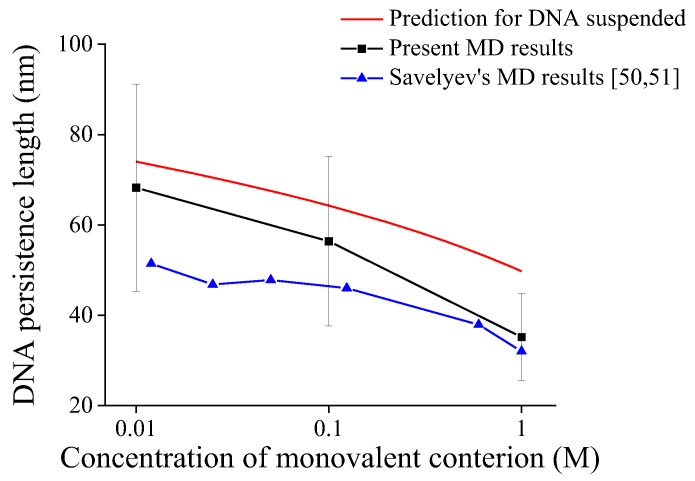
Comparison of predictions and MD results for persistence length of DNA suspended in solution.

**Figure 4 molecules-27-07769-f004:**
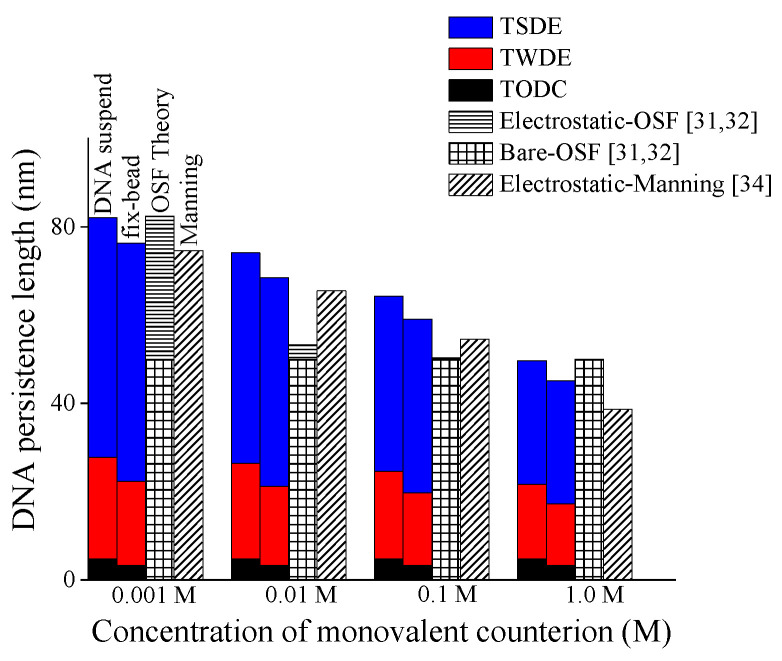
Respective contributions of three addictive terms (TODC, TWDE, TSDE) to persistence length of DNA with different end-constraints versus monovalent salt concentration.

**Figure 5 molecules-27-07769-f005:**
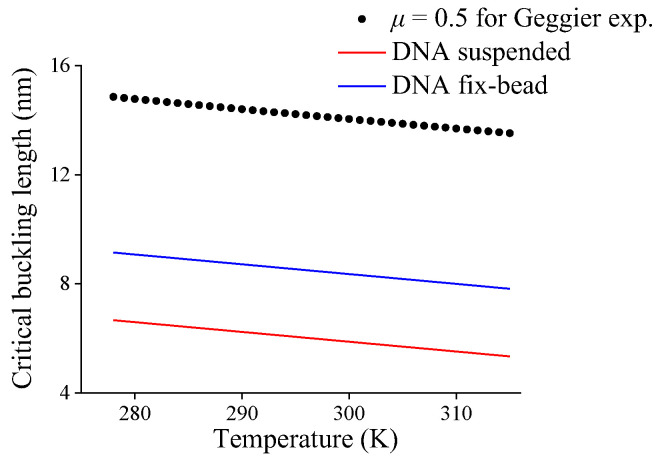
Critical buckling length of DNA fragment with different end-constraints versus temperature.

**Figure 6 molecules-27-07769-f006:**
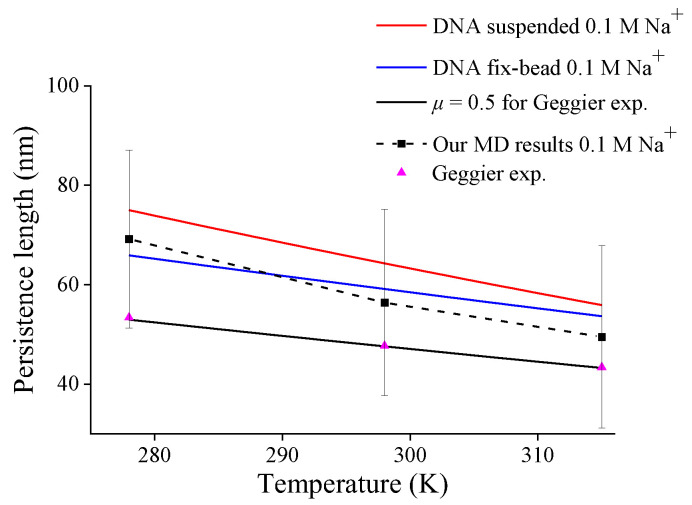
DNA persistence length with different end-constraint conditions versus temperature.

**Figure 7 molecules-27-07769-f007:**
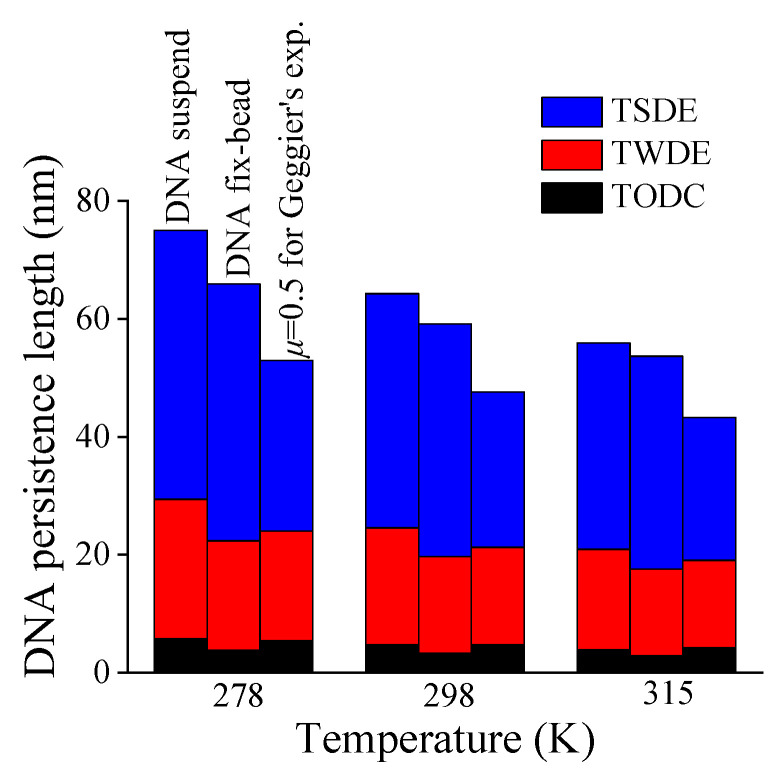
Contributions of three additive terms (TODC, TWDE, TSDE) to DNA persistence length versus temperature.

**Table 1 molecules-27-07769-t001:** Summary of experiments measuring DNA persistence length and corresponding fitted parameters.

Experiment Method	DNA Constrained States	Temperature K	Kind of Salt	Concentrations of Salt mol/L	DNA Contour Length bp	DNA Persistence Length nm	Fitted Effective Length Factor *μ*	Fitted Critical Buckling Length *l*_cb_ nm	Goodness of Fit
Rayleigh light scattering [16]	Suspended in solution	293	NaH_2_PO_4_	0.005–3	39,000	38–78	0.668	9.16	0.893
Linear dichroism [49]	Suspended in solution	298	NaCl	0.002–1	39,000	40.2–66.9	1.256	4.18	0.652
Flow birefringence [17]	Suspended in solution	298	NaCl	0.002–1	39,000	39.9–72.3	1.065	5.14	0.815
Dynamic light scattering [15]	Suspended in solution	298	NaCl	0.005–4	6594	32.0–91.4	1.095	5.31	0.761
Optical tweezers [10]	Fix-bead	298	NaCl	0.00186–0.586	48,500	45.1–96.3	0.635	9.64	0.511
Tethered-particle method [9]	Fix-bead	298	NaCl	0.015–3.33	2060	34.9–54.7	0.835	6.52	0.824
Tethered-particle method [9]	Fix-bead	298	NaCl	0.015–3.33	1201	31.3–52.2	0.686	7.88	0.911
Tethered-particle method [9]	Fix-bead	298	MgCl_2_	0.003–0.104	2060	37.2–50.1	0.736	8.95	0.614
Tethered-particle method [9]	Fix-bead	298	MgCl_2_	0.003–0.104	1201	36.7–46.1	0.700	9.18	0.624
Electron microscopy [48]	Adsorbed on substrate	298	NaCl	0.03–0.5	4800	54.5–145.6	0.489	15.83	0.404

**Table 2 molecules-27-07769-t002:** Effective length factors and critical buckling lengths for DNAs suspended in solution or with fix-bead constraint.

DNA Constrained States	Effective Length Factors *μ* ± Standard Deviation	Critical Buckling Length *l*_cb_ ± Standard Deviation
Suspended in solution	1.021 ± 0.250	5.95 ± 2.20
Fix-bead constraint	0.718 ± 0.075	8.43 ± 1.25

## Data Availability

All data is contained within the article or Appendix A.

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
