# Peer review of "Mechanical Constraint Effect on DNA Persistence Length"

_molecules, 2022, doi:10.3390/molecules27227769_

Round 1
Reviewer 1 Report
This is an interesting paper dealing with the nature of the DNA persistence length. It combines some previous ideas, mostly based on Manning's papers, and derives an alternative relationship between the value of the persistence length and salt concentration for variously constrained chains. The new theory could be better then the well known ones, but the quality of data does not really allow a definitive answer in my opinion. I think the paper should be published but I would suggest the authors do the following minor improvements in the revised version of the paper:
- for a paper dealing with the details of the DNA persistence length it is appropriate to cite the first measurement of this quantity which is A. Peterlin, Light Scattering by very Stiff Chain Molecules, Nature, 171 259-260 (1953). (same year as the Watson-Crick paper)
- the properties of the DNA persistenc length are intimately connected with another elastic modulus: the stretching modulus, which also varies wit the ionic strength and which the authors do not specifically cover or include in their theory. I suggest they cite another work where this is done in detail: Podgornik, Hansen, and Parsegian, Elastic moduli renormalization in self-interacting stretchable polyelectrolytes, J. Chem. Phys. 113 (2000)9343-9350.
Author Response
- Point 1: - for a paper dealing with the details of the DNA persistence length it is appropriate to cite the first measurement of this quantity which is A. Peterlin, Light Scattering by very Stiff Chain Molecules, Nature, 171 259-260 (1953). (same year as the Watson-Crick paper)
Response:We agree with the reviewer’s comment. We have cited this paper in revised manuscript.
- Point 2: - the properties of the DNA persistence length are intimately connected with another elastic modulus: the stretching modulus, which also varies with the ionic strength and which the authors do not specifically cover or include in their theory. I suggest they cite another work where this is done in detail: Podgornik, Hansen, and Parsegian, Elastic moduli renormalization in self-interacting stretchable polyelectrolytes, J. Chem. Phys. 113 (2000)9343-9350.
Response:We agree with the reviewer’s comment and this paper have been introduced and cited in revised manuscript.

Reviewer 2 Report
In this manuscript, the authors applied buckling theory, electrostatic force and thermal fluctuation to develop a mathematical model of DNA to explain the mechanical property of constrained DNA molecules in various experimental conditions. They derived the relationship between the persistence and buckling length of a DNA. Then they used the equation to fit the buckling length of DNA from previous literatures. The same equation was used to calculate the persistence length of DNA in different concentrations of monovalent ions which showed good agreement with the experimental data. The ion dependency was further validated by comparing the result with MD simulations. Finally, contribution of each component for persistence length of DNA was analysed.
Overall, I think this paper adds an interesting viewpoint on the mechanical properties of DNA. The calculations show qualitatively good agreements with the previous experimental data and MD results. Also, the analytical solution allows authors to separate the contributions from each term to the persistence length of DNA at different salt concentrations. I believe the manuscript can be published after some of my minor comments are addressed.
The authors used GC-rich sequence for their MD simulation. Could you provide a reason for that?
The average for the buckling length was presented but the error was not. How the error in the buckling length would change the result?
The goodness of the fitting seems to be different for each experiment. Do authors take account of this when averaging the buckling length?
In the Table 2, a unit(nm) is for the length is missing
Author Response
- Piont 1: The authors used GC-rich sequence for their MD simulation. Could you provide a reason for that?
Response: Our MD simulations is inspired by the work of Wu et al. (Reference 44 in revised manuscript), who used MD simulations to study finite-length effects on flexibility of short DNA, and we selected the same DNA sequence as their simulations to study environmental effects on DNA persistence length. Moreover, as reported by Wu et al., this DNA sequence was derived from an experiment in which the mean and variance of DNA end-to-end length are measured using small-angle X-ray scattering [1]. We have added the detail for selecting this DNA sequence in revised manuscript.
- Point 2: The average for the buckling length was presented but the error was not. How the error in the buckling length would change the result?
Response: We agree with the reviewer’s comment, the error of average buckling length does exist. It is necessary to note that the end-constraint is one cause of buckling and the critical buckling length is one result in mechanical mechanism, and this causality is implicitly expressed in Eq. (4) with effective length factor μ and critical buckling length lcb. Because both parameters are unknown, we obtain them by fitting model with data of experiments and process statistical average for them. On the surface, the average values of both parameters will have errors, the error of buckling length actually is the result caused by that of effective length factor because of the implicit causality. If the errors of both parameters are considered when both parameters are substituted into Eq. (7) to predict DNA persistence length, the errors from the same cause are overlapped in calculation actually, resulting in excessively amplifying or reducing the prediction. Therefore, we only consider the error of effective length factor but discard the error of buckling length in predicting DNA persistence length. We have added the standard deviation for average buckling length in Table 2 and made necessary explanations in revised manuscript.
- Point 3: The goodness of the fitting seems to be different for each experiment. Do authors take account of this when averaging the buckling length?
Response: We have excluded the results with much low goodness of fit, the most of the fitted results have goodness of fit in a concentrated range around 0.6 to 0.8, so the averaging fitted buckling length is reliable. Moreover, it is a valuable suggestion, and we can take account of the goodness of fit for averaging in future work.
- Point 4: In the Table 2, a unit(nm) is for the length is missing
Response: We have added unit for the buckling length in Table 2.
[1] Mathew-Fenn, R. S.; Das, R.; Harbury, P. A. B., Remeasuring the double helix. Science 2008, 322, (5900), 446–449.

Reviewer 3 Report
The authors here present an elegant theoretical model to describe the persistence length of double stranded DNA. The study adds a valuable contribution to theoretical understanding of this parameter which is still an important subject of debate within the field. Especially the experimental observation of the comparably soft behavior on very small length scales is of great interest for both DNA nanotechnology as well as for physiological processes like the binding of DNA to histones and had already previously been attributed to buckling of DNA. By creating a surprisingly simple model, treating DNA as an elastic cylindrical rod and combining Euler’s buckling theory with electrostatic model conceptions introduced by Manning, they are able to fit experimental data from literature obtained at a large range of parameters (buffer conditions) to their model. Further they validated their results with their own molecular dynamics simulation of a short DNA rod. Considering the manuscripts relevance for the field and the quality of the research, I can recommend the publication in molecules if further comments are addressed. The overall text is well written and understandable, however some passages the language could be slightly improved (I noted only some examples, however, I’m not a native English speaker myself)
Further comments:
Key insights and determined parameters could be made more accessible readers with less theoretical background (DNA nanotechnologists) by highlighting those a little more prominently in the conclusion.
The authors summarize various different theoretical models that have been used to describe DNA persistence length. I wonder if it would be possible to provide a direct comparison of these different models for example within a table or figure and thereby set their model in context to existing theory.
The model treats DNA very crudely as a semi rigid rod. While this model seems to fit literature values surprisingly well, I would like to know if it is possible to make any statements about the range where this crude model is still applicable. Do the authors have an idea upon which point the finer structure for example base pairing and frying or sequence specific motives like A-tracks can play a role and have to be considered?
Line 43: persistence length is not defined as 50nm in a mathematical sense. Rather it is frequently approximated in literature with about 50 nm at standard conditions.
Line 87-90: Here theoretical considerations of Manning are discussed. A bit more context to explain his concept better would be helpful.
Line 137 is making a quite trivial statement in a relatively complicated way (to my opinion) and therefore I’m not sure if reference 34 and 35 is necessary. But maybe I just do not see the deeper insight here.
Line 187-197: The definition of the effective length factor and its physical meaning is not clear from this passage.
Line 247-251: the wording of this passage is quite difficult to understand for me
Table 1 compares experiments measuring “DNA persistence length”, presenting different experimental conditions as well as results. However no persistence length values are actually provided. Considering that the authors state that these values differ quite a lot for different studies, it might be interesting to integrate these values into the table. If these studies provide a range of values for different tested condition, exemplary comparable values or value ranges might be able to state.
Also, con the “goodness of fit” parameter be defined or specified more?
I wonder how representative 50 different conformations collected for each configurations of the simulated DNA duplex are. If possible, adding standard deviation values (or any other measure for the accuracy of the determined value) to the corresponding graphs of the determined persistence length graphs would be valuable.
Author Response
- Point 1: Key insights and determined parameters could be made more accessible readers with less theoretical background (DNA nanotechnologists) by highlighting those a little more prominently in the conclusion.
Response: We agree with the reviewer’s comment. We have adjusted some statement and highlighted the value of effective length factor for different experiments to make it easier to understand in Conclusion of revised manuscript.
- Point 2: The authors summarize various different theoretical models that have been used to describe DNA persistence length. I wonder if it would be possible to provide a direct comparison of these different models for example within a table or figure and thereby set their model in context to existing theory.
Response: In fact, we have compared our theoretical model with relevant theoretical models in some figures, for examples, in Fig.2 and Fig.4 we have compared the predictions by our model with those by Manning’s and OSF theoretical models for ionic depended DNA persistence length. While in Fig.6, we have compared our model with Geggier’s empirical expression for temperature depended DNA persistence length. However, not all theoretical models mentioned can be directly compared, some theoretical models aimed to investigate effects of other factors on DNA persistence length, for example, Seol et al.’s theoretical model revealed the dependence of DNA persistence length on DNA contour length but not on ionic conditions and temperature, so the comparison of their results and ours cannot be shown in a table or figure.
- Point 3: The model treats DNA very crudely as a semi rigid rod. While this model seems to fit literature values surprisingly well, I would like to know if it is possible to make any statements about the range where this crude model is still applicable. Do the authors have an idea upon which point the finer structure for example base pairing and frying or sequence specific motives like A-tracks can play a role and have to be considered?
Response: Our model is multiscale, which means that the DNA with length of tens of nanometers exhibits macroscopic mechanical properties, and mesoscopic interactions, such as electrostatic forces and thermal fluctuation, can exert influence on the DNA. We have coarse-grained DNA as a homogeneous material, and the studied macroscopic mechanical properties and the used mesoscopic interactions models did not reflect the effect at the molecule level of base pair. It will be a worthwhile work by more refined models to study dependences of base pair and sequence on DNA properties in the future. We have added contexts about the applicable range in revised manuscript.
- Point 4: Line 43: persistence length is not defined as 50 nm in a mathematical sense. Rather it is frequently approximated in literature with about 50 nm at standard conditions.
Response: We agree with the reviewer’s comment. We have adjusted the statement in revised manuscript.
- Point 5: Line 87-90: Here theoretical considerations of Manning are discussed. A bit more context to explain his concept better would be helpful.
Response: We agree with the reviewer’s comment. We have added more contexts about Manning’s theory and concept in revised manuscript.
- Point 6: Line 137 is making a quite trivial statement in a relatively complicated way (to my opinion) and therefore I’m not sure if reference 34 and 35 is necessary. But maybe I just do not see the deeper insight here.
Response: The statement noted by reviewer is located in lines 141-143 of the original manuscript. The concept of “Biological shape is revealing of the physical forces” is taken from reference 34 and 35. We have adjusted the statement concisely to make it easier to understand.
- Point 7: Line 187-197: The definition of the effective length factor and its physical meaning is not clear from this passage.
Response: In Supplementary Information I, we have derived the critical buckling length of buckling rod with any elastic end-constraint, and degenerate the critical buckling length to expression with specific hinge-hinge end-constraint. A parameter μ named effective length factor in buckling research field is defined as a ratio of the critical buckling length with hinge-hinge end-constraint to that with any elastic end-constraint [1]. To be formally consistent with the traditional Euler’s formula, the critical buckling length with elastic end-constraint is transformed into expression with effective length factor μ. We have added more contexts about effective length factor and cited the relevant reference in revised manuscript and revised Supplementary.
- Point 8: Line 247-251: the wording of this passage is quite difficult to understand for me.
Response: We have adjusted the statement in revised manuscript to make it easier to understand.
- Point 9: Table 1 compares experiments measuring “DNA persistence length”, presenting different experimental conditions as well as results. However no persistence length values are actually provided. Considering that the authors state that these values differ quite a lot for different studies, it might be interesting to integrate these values into the table. If these studies provide a range of values for different tested condition, exemplary comparable values or value ranges might be able to state.
Response: We agree with the reviewer’s comment. We have added the range of DNA persistence length measured by different experiments in Table 1 of revised manuscript.
- Point 10: Also, can the “goodness of fit” parameter be defined or specified more?
Response: The goodness of fit describes how well a predicted model fits a set of measurements, and the coefficient of determination is a general parameter for quantifying the goodness of fit, and we also used this criterion in our work. The most common definition of the coefficient of determination denoted r2 is expressed as r2=1–SSR/TSS, where SSR is the sum of squares of residuals and TSS is the total sum of squares. In the most idealized fitting case, the predicted values by model exactly match the measured value, which results in SSR=0 and r2=1. We have added more context defining goodness of fit in revised manuscript.
- Point 11: I wonder how representative 50 different conformations collected for each configurations of the simulated DNA duplex are. If possible, adding standard deviation values (or any other measure for the accuracy of the determined value) to the corresponding graphs of the determined persistence length graphs would be valuable.
Response: We agree with the reviewer’s comment. We have added some figures of MD simulated DNA conformations in Supplementary Information III, and have added the standard deviation of DNA persistence length by MD simulations in Fig.3 and Fig.6 of revised manuscript.
[1] Timoshenko, S. P.; Gere, J. M., Theory of Elastic Stability: Second Edition. Dover Publications, Inc.: New York, 1989.
